# Impact of Exogenous Nitric Oxide Treatment on Vascularization of a Subcutaneous Device for Cell Transplantation

Alexandra M. Smink [1,2,*], Bryan Ceballos [2], Taco Koster [1], Samuel Rodriquez [2], Michael Alexander [2], Jonathan R. T. Lakey [2,3] and Paul de Vos [1]

1   Department of Pathology and Medical Biology, University Medical Center Groningen, University of Groningen, 9713 Groningen, The Netherlands
2   Department of Surgery, University of California Irvine, Orange, CA 92697, USA
3   Department of Biomedical Engineering, University of California Irvine, Irvine, CA 92602, USA
*   Correspondence: a.m.smink@umcg.nl; Tel.: +31-50 3617865

**Abstract:** Subcutaneous polymer scaffolds have shown potential for creating an optimal transplantation site in cellular replacement therapy, e.g., when transplanting insulin-producing cells to cure type 1 diabetes. Imperative for these scaffolds is a high degree of vascularization to guarantee long-term functional cellular survival. In this study, the effect of the nitric oxide (NO) donor *S*-nitroso-*N*-acetyl-DL-penicillamine (SNAP) on the vascularization degree of a subcutaneous poly(D,L-lactide-co-ε-caprolactone) (PDLLCL) scaffold was investigated. To this end, scaffolds were implanted under the skin of C57BL/6 mice. Each mouse received a control scaffold and a scaffold containing SNAP. At day 7, 14, and 28, the oxygen percentage within the scaffolds was measured and at day 28, the vascularization degree was determined with lectin infusion and gene expression analysis. We measured lower oxygen percentages within the scaffolds containing the NO-donor up to day 14 compared to the control scaffolds, but no differences were found at day 28. Although blood vessels in the scaffolds were well perfused, no differences between the groups were found in the lectin staining and gene expression of vascular markers, such as CD31, CD105, and VEGFa. To conclude, in this biomaterial setting, addition of a NO-donor did not improve the vascularization degree of the subcutaneous scaffold.

**Keywords:** nitric oxide; subcutaneous; vascularization; polymer scaffold

## 1. Introduction

Cellular replacement therapy has emerged as a treatment option for several diseases, such as chronic liver diseases [1,2], retinal degeneration [3,4], neural tract damage [5,6], and type 1 diabetes [7,8]. By transplanting healthy cells into the patients, the non-functional diseased cells are replaced. The success of cellular replacement therapy depends on the successful delivery of the cells, and this often impedes clinical application. To this end, cellular delivery devices or so-called transplantation scaffolds are developed. Such a scaffold allows the targeted delivery of cells and supports engraftment into the host tissue. These scaffolds are developed for transplanting adult cells, such as when transplanting the insulin-producing pancreatic islets to treat type 1 diabetes [9,10], but are even more important when transplanting stem cells. Stem cell-derived insulin-producing cells for example still have limitations in function and might exhibit impairments such as abnormal hormone release or teratoma formation [11]. A scaffold provides a confined space that allows fast and safe retrievability in case of complications.

The subcutaneous site seems to be a promising site for cell transplantation since it is easily accessible via a minimally invasive procedure, allows non-invasive monitoring of function, is retrievable in case of graft failure or need for replacement, and there is enough volume. However, the unmodified skin does not provide an environment that is suitable

for transplanted cells, since the graft depends on the diffusion of oxygen and nutrients during the immediate post-transplantation period, while the revascularization under the skin is poor [12]. Fast graft revascularization and the supply of oxygen and nutrients are imperative for the survival of cells and their function, e.g., release of hormones or repair of damage. For instance, 50% of transplanted islets are lost during this immediate post-transplant period due to hypoxia because of incomplete and slow graft revascularization [13–15]. Therefore, a subcutaneous transplantation scaffold should not only provide three-dimensional support to the cells but should also enhance vascularization [9,16,17].

Angiogenesis is a complex biological process to form new blood vessels from pre-existing vessels. One of the contributing factors to this process is the gasotransmitter nitric oxide (NO) [18]. Downstream from the angiogenic pathway stimulated by growth factors, such as VEGF, NO significantly contributes to blood vessel formation by promoting endothelial cell migration and differentiation [18,19]. Blocking NO results in the attenuation of vascularization [20]. The NO pathway, therefore, appears to be a promising target for a therapeutic strategy to improve the vascularization of subcutaneous scaffolds. However, the half-life time of NO is extremely short—it is in the range of several seconds depending on the concentration and the surrounding environment [21]. Therefore, the range of action is limited to 100–200 μm. These characteristics make it challenging to deliver NO at the therapeutic site and so far, the number of NO-release systems is limited.

The NO-donor *S*-nitroso-*N*-acetyl-DL-penicillamine (SNAP) is considered minimally toxic, and it does not induce oxidative and nitrosative stress [21]. Previous studies have shown that SNAP can provide local and controlled release of NO from synthetic materials [22,23]. Furthermore, SNAP has been reported to induce endothelial tube formation in an in vitro angiogenesis assay and in vivo angiogenesis in subcutaneously implanted Matrigel [19]. Here, we investigate in mice if the addition of the NO-donor SNAP to a subcutaneous polymer scaffold improves its vascularization degree. This could lead to improved cell transplantation outcomes. However, in our biomaterial setting, the addition of NO did not lead to improved vascularization compared to the control scaffolds.

## 2. Materials and Methods

### 2.1. Scaffold Fabrication

Porous poly(D,L-lactide-co-ε-caprolactone) (PDLLCL; Sigma-Aldrich, Zwijndrecht, The Netherlands) scaffolds (1 × 1.5 × 0.5 cm) were manufactured via salt leaching, as previously described [9]. Briefly, sodium chloride (particle size 250–425 μm; Sigma-Aldrich) was added to the PDLLCL/chloroform solution (4% *w/v*; Sigma-Aldrich, Burlington, MA, USA) in a ratio of 10:1 *w/w*. After complete evaporation of the chloroform, the sodium chloride particles were thoroughly washed from the PDLLCL sheet with sterile water. The polymer sheet was resized into scaffolds of 1 by 1.5 cm and sterilized with 70% ethanol (Merck, Rahway, NJ, USA; Sigma-Aldrich).

### 2.2. Scaffold Implantation

On the day of implantation, 2 mg/mL of fibrinogen (Millipore, Temecula, CA, USA) was mixed with the NO-donor *S*-Nitroso-*N*-acetyl-DL-penicillamine (SNAP, 35 mg/mL; Sigma-Aldrich). Before transferring the solution to the scaffold, 100 U/mL thrombin IIa (Millipore) was added in order to initiate crosslinking within the pores of the scaffold. Fibrinogen without the NO-donor was used for the control scaffolds. Scaffolds were subcutaneously implanted on the back of male C57BL/6 mice (Charles River, Wilmington, DE, USA). Each mouse received a control scaffold without the NO-donor and a scaffold with the NO-donor (n = 3). Scaffolds were placed 4 cm apart from each other in separate subcutaneous pockets to prevent interaction. After surgery, all mice received ibuprofen water (Banner Pharmacaps, High Point, NC, USA; 0.2 mg/mL) for two days. Animals were housed at the University of California Irvine animal facility and maintained under 12 h light/dark cycles with ad libitum access to water and standard chow. This study was

conducted with the approval of The University of California Institutional Animal Care and Use Committee at the University of Irvine (IACUC # 2008-2850).

### 2.3. Blood Vessel Functionality

On days 7, 14, and 28 after implantation, the percentage of oxygen within the scaffolds was measured with the Microx 4 precision sensing system (PreSens, Regensburg, Germany). To this end, the oxygen probe was placed in a minimally invasive manner in the scaffold for 10 min while the animal was anesthetized. Furthermore, to determine the perfusion of the blood vessels within the scaffolds, the mice were perfused with DyLight 649 labeled Lycopersicon Esculentum (Tomato) lectin (Vector Laboratories; Brunschwig Chemie, Amsterdam, The Netherlands) on the day of sacrifice (day 28). Briefly, via the penile vein, 200 µL lectin was injected under anesthesia. After 15 min of perfusion, the vasculature was flushed with a saline injection into the heart and the scaffolds were explanted. Half of the explanted scaffold was fixed in 2% paraformaldehyde to image the lectin staining by confocal microscopy (Leica SP8; Leica Microsystems B.V., Amsterdam, The Netherlands). Lectin images were analyzed with the skeleton macro plugin of Image J (Version 2.0.0-rc-69/1.52p; National Institutes of Health, Bethesda, MD, USA) as described by Weaver et al. [24].

### 2.4. Reverse Transcription Polymerase Chain Reaction (RT-PCR)

The mRNA expression of several vascularization markers was measured within the other half of the explanted scaffold with real-time RT-PCR. RNA was extracted using Trizol according to the manufacturer's protocol (Invitrogen; Thermo Scientific, Landsmeer, The Netherlands). Possible DNA contaminants were removed by the DNA-freeTM DNA removal kit (Invitrogen, Waltham, MA, USA). The SuperScript® III Reverse Transcriptase kit (Life Technologies; Thermo Scientific) was used for cDNA synthesis. Primer and probe sets of the vascularization genes CD31, VEGFa, VE-Cadherin, CD105, angiopoietin 1 and 2 for TaqMan Gene Expression Assays were purchased from Thermo Scientific (Table 1). The ViiA^TM Real Time PCR system (Life Technologies, Foster City, CA, USA) was programmed as follows: subsequently 2 min at 50 °C, 10 min at 95 °C, 15 s at 95 °C, and 60 s at 60 °C, while repeating the last two steps for 40 cycles. Delta Ct values were calculated and normalized against the expression of the housekeeping gene GAPDH. Delta-delta Ct values were used to determine the fold change in the NO-treated scaffold compared to the control scaffold.

**Table 1.** TaqMan assay IDs of RT-PCR genes (Thermo Scientific).

| Gene | Assay ID |
|---|---|
| GAPDH | Mm99999915_g1 |
| CD31 (PECAM1) | Mm01242576_m1 |
| VEGFa | Mm00437306_m1 |
| VE-Cadherin (CDH5) | Mm00486938_m1 |
| CD105 (Endoglin) | Mm00468252_m1 |
| Angiopoietin 1 | Mm00456503_m1 |
| Angiopoietin 2 | Mm00545822_m1 |

### 2.5. Statistics

Statistical analysis was carried out in GraphPad Prism (version 8.4.0; GraphPad Software, Inc., La Jolla, CA, USA). A Shapiro–Wilk normality test was performed to test the data for normality. To test differences between the groups, a two-way ANOVA was applied for the oxygen measurements and a Mann–Whitney test was used for the lectin and PCR data, and $p$-values $< 0.05$ were considered significant. The data are presented in mean $\pm$ standard error of the mean.

## 3. Results

### 3.1. NO Impairs the Oxygenation during the Vascularization Process

To follow the vascularization process of the subcutaneous scaffolds over time in vivo in a minimally invasive manner, the oxygen percentage was measured on days 7, 14, and 28 (Figure 1). The oxygen percentage within the control and NO-donor scaffolds at day 7 was, respectively, 16.6% ± 1.3% and 16.0% ± 3.0%. On day 14, a difference was observed between the oxygen percentages of these groups. The control group showed an oxygen percentage of 18.4% ± 1.0%, whereas the NO-donor-containing scaffolds had a lower oxygen percentage of 15.2% ± 1.2%. This difference was not observed on day 28, when the oxygen percentage of the control and NO groups were, respectively, 17.9% ± 0.4% and 18.3% ± 1.4. Overall, NO treatment had a statically significant effect on oxygen percentage within the scaffold ($p < 0.05$), but no significant differences were obtained at the individual time points. Therefore, it seems that NO treatment impaired the vascularization process, but did not influence the final degree of oxygenation of the device, which was similar between the NO-donor and control scaffolds.

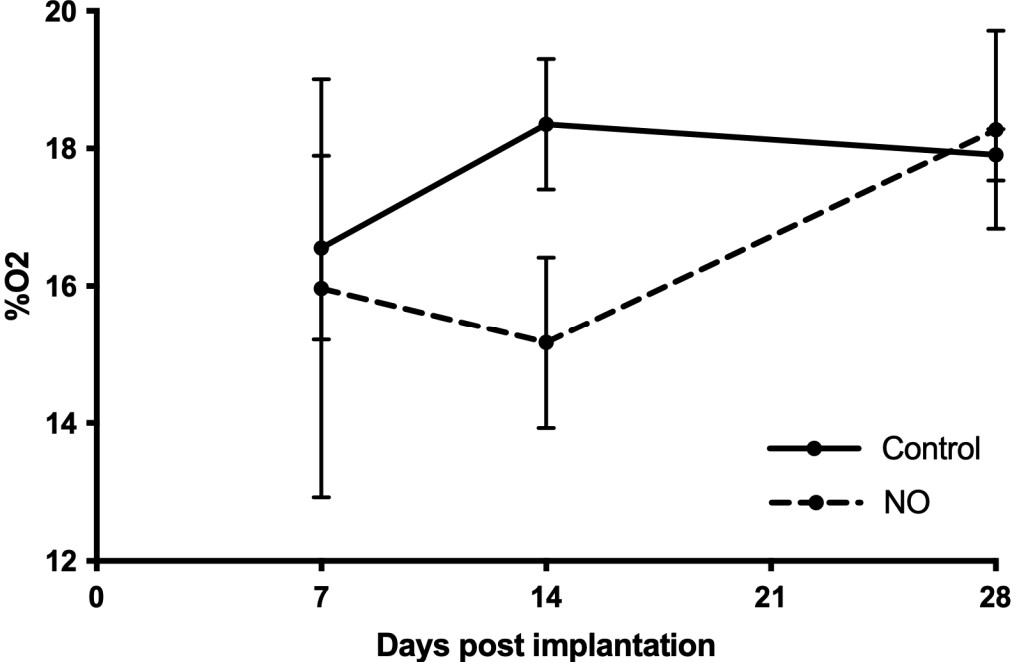

**Figure 1.** Oxygen percentage within subcutaneous scaffolds. After implantation of the control scaffold and the scaffold containing the NO-donor, the oxygen percentage was measured on day 7, 14, and 28 with the Microx 4 PreSens system. The mean and standard error of the mean are plotted (n = 3); statistical analysis was carried out using a two-way ANOVA with a Bonferroni post-hoc test, $p < 0.05$. This test indicated a significant treatment effect and no time effect.

### 3.2. Blood Vessels Are Well Perfused

Lectin perfusion on the day of sacrifice showed that scaffolds of both groups are well perfused (Figure 2). However, no differences were observed when analyzing the lectin staining between the control group and the NO-treated group. The addition of the NO-donor SNAP to the scaffold did not improve the number of vessel branches or junctions, and nor did the average and maximal branch length of the vessels change.

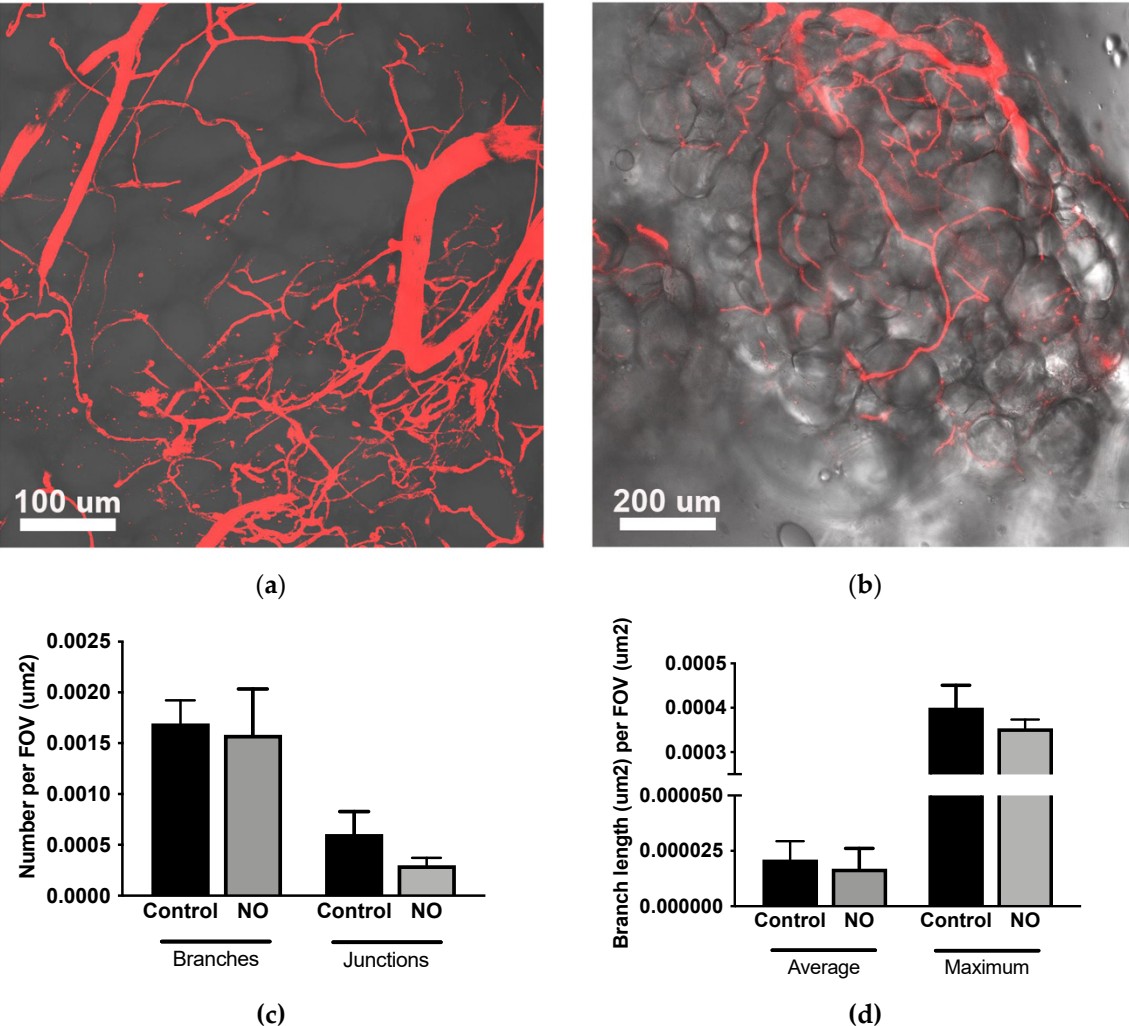

**Figure 2.** Lectin staining of scaffolds after 28 days. Confocal images from the lectin staining of control (**a**) and NO-treated (**b**) scaffolds. Confocal pictures were quantified by the skeleton macro plugin of Image J (**c**,**d**). The mean and standard error of the mean are plotted (n = 3); statistical analysis was carried out using a Mann–Whitney test, $p < 0.05$.

### 3.3. NO Treatment Does Not Increase mRNA Expression of Vascularization Genes

Part of the scaffold was processed for PCR to investigate the mRNA expression of several vascularization genes. These results confirm the lectin-vessel perfusion analysis (Figure 3). No statistically significant differences were found between the control group and the NO-treated group. The fold change in the endothelial marker CD31 in the NO group was $1.8 \pm 0.6$ compared to the control group ($1.0 \pm 0.06$). The fold change in the angiogenesis marker CD105 in the NO group was $1.6 \pm 0.06$, compared to $1.1 \pm 0.3$ for the control group. Similar results were obtained for the mRNA expression of VEGFa, VE-cadherin, angiopoietin 1, and angiopoietin 2 (Figure 3c–f). This indicates that the NO-donor did not improve the vascularization of the subcutaneous polymer device.

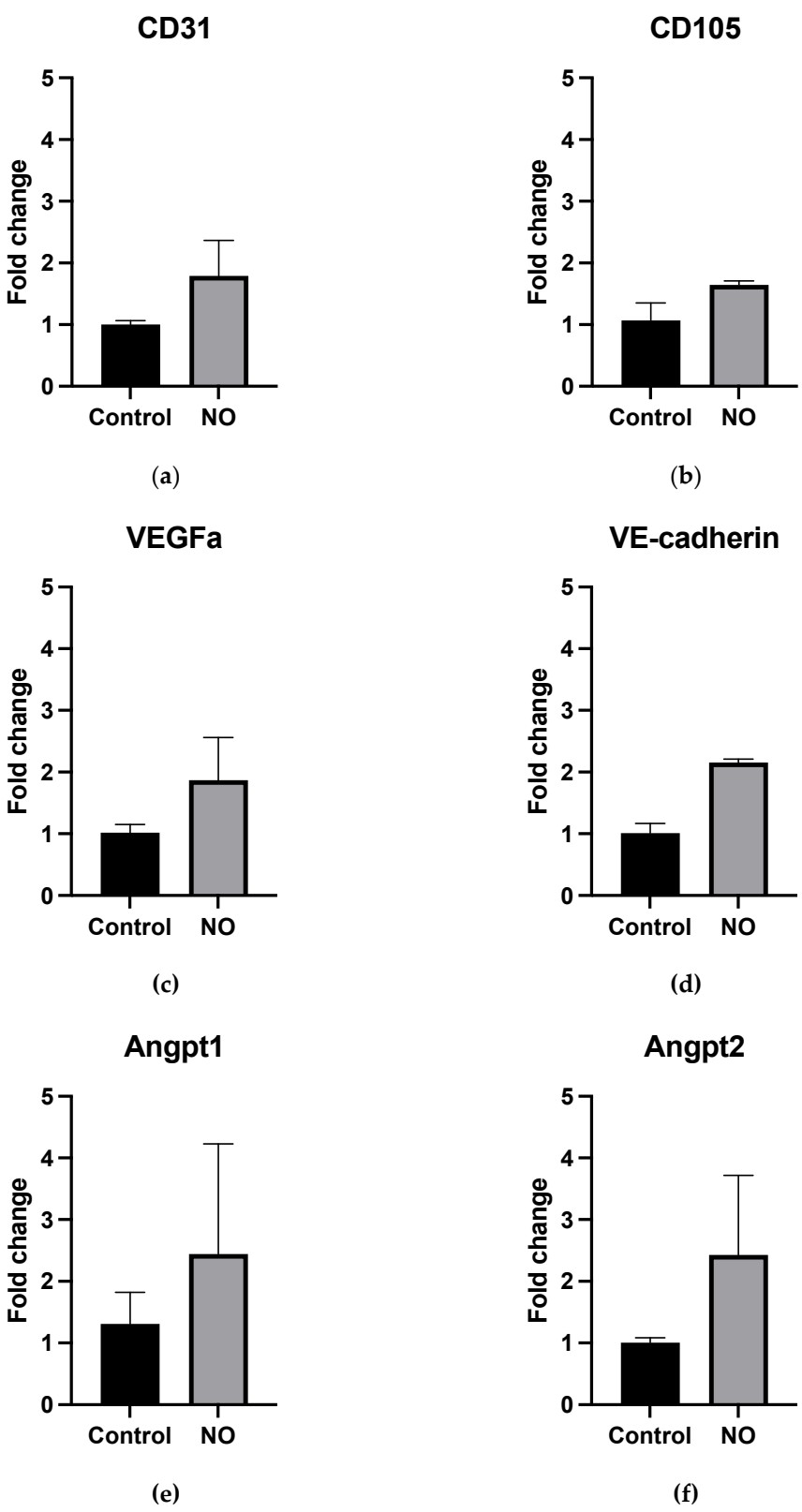

**Figure 3.** mRNA expression of vascularization markers within the scaffolds after 28 days. The gene expression of the endothelial cell marker CD31 (**a**), angiogenesis marker CD105 (**b**), vascular growth factor VEGFa (**c**), VE-cadherin (**d**), and angiopoietin 1 (**e**) and 2 (**f**) measured by RT-PCR. The mean and standard error of the mean are plotted (n = 3); statistical analysis was carried out using a Mann–Whitney test, $p < 0.05$.

## 4. Discussion

The formation of a vascular network within a subcutaneous scaffold for cell transplantation requires the activation of endothelial cells, their migration, and proliferation. NO has been shown to contribute to angiogenesis as a key signaling molecule and regulator [19,25]. Accordingly, we examined the angiogenic effects of exogenous NO treatment on the vascularization of a scaffold. Here we show that the addition of the NO-donor SNAP did not improve the vascularization of our PDLLCL scaffold under the skin. No differences were found between control scaffolds and NO scaffolds in the mRNA expression of vascular markers or the number of blood vessels. The oxygen perfusion was even significantly reduced during the one-month implantation in the NO-treated scaffolds.

Previous in vitro studies have shown that coating biomaterials with the NO-donor SNAP resulted in increased endothelial migration, endothelial proliferation, and protection against bacteria [22,26]. Furthermore, NO coating of vascular stents increased in vivo the binding and growth of endothelial cells on the outside of the stent, whereas inside, the hemocompatibility was also improved by there being less binding of platelets [27,28]. Lee et al. [19] even showed increased angiogenesis in Matrigel plugs with SNAP implanted under the skin. However, here we did not find a positive effect of NO on vascularization. The effect of NO seems to vary depending on its concentration [21,29]. Therefore, the concentration of NO and the timing of the release are imperative for achieving the desired effect.

Since it is known that NO diminishes the foreign body response [30], and this response is involved in the vascularization of our type of scaffold [31], it might be suggested that NO inhibits the foreign body response against the scaffold and subsequently attenuates the vascularization. Previous studies using different NO-release systems have shown that there is first a burst release of NO, followed by a steady-state release in the physiological range up to 15 days [23,26]. This could explain why we observed a decrease in oxygen percentage up to day 14, but not on day 28. The NO-donor was exhausted after 15 days and could therefore no longer inhibit the foreign body response enabling the vascularization to recover to similar levels as the control. Because of this, no differences between the NO and control group were found in oxygen percentage, the number of blood vessels, and mRNA expression of vascular markers at the end of the study.

In conclusion, NO did not improve the vascularization of the subcutaneous PDLLCL scaffold. This indicates that in this biomaterial setting, the addition of this NO-donor will not improve cellular transplantation. The tissue engineering field should be aware that the optimal NO release kinetics and doses remain to be elucidated for each biomaterial and implantation site due to the broad range of NO activities.

**Author Contributions:** Conceptualization, A.M.S., J.R.T.L., and P.d.V.; methodology, A.M.S., M.A., J.R.T.L., and P.d.V.; software, not applicable; validation, A.M.S. and P.d.V.; formal analysis, A.M.S.; investigation, A.M.S., B.C., T.K., S.R., and M.A.; resources, A.M.S. and M.A.; data curation, A.M.S.; writing—original draft preparation, A.M.S.; writing—review and editing, A.M.S., B.C., T.K., S.R., M.A., J.R.T.L., and P.d.V.; visualization, A.M.S.; supervision, A.M.S.; project administration, A.M.S.; funding acquisition, A.M.S. and P.d.V. All authors have read and agreed to the published version of the manuscript.

**Funding:** This research was funded by the Juvenile Diabetes Research Foundation (JDRF), research grant 3-PDF-2018-594-A-N.

**Institutional Review Board Statement:** The animal study protocol was approved by the Institutional Review Board (and Ethics Committee) of the University of California, Irvine (IACUC # 2008-2850).

**Informed Consent Statement:** Not applicable.

**Data Availability Statement:** The datasets generated during and analyzed during the current study are available from the corresponding author upon request.

**Acknowledgments:** The authors thank Sarah Lee, Shiri Li (Department of Surgery, University of California, Irvine), and Bart de Haan (Department of Pathology and Medical Biology, University Medical Center Groningen) for their technical assistance and support during this study.

**Conflicts of Interest:** The authors declare no conflict of interest. The funder had no role in the design of the study; in the collection, analyses, or interpretation of data; in the writing of the manuscript; or in the decision to publish the results.

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
