# Peer review of "Impact of Exogenous Nitric Oxide Treatment on Vascularization of a Subcutaneous Device for Cell Transplantation"

_2673-6209, doi:10.3390/macromol2030029_

Round 1

Reviewer 1 Report

The study is devoted to the actual problem of the effect of exogenous nitric oxide on the vascularization of the subcutaneous scaffold. A detailed, reasoned and honest study with a negative result, which will be very useful.

Author Response

We thank the reviewer for finding our study interesting for the research community.

Reviewer 2 Report

The problem of vascularisation and cell survival has not yet been solved in tissue engineering and cell therapies. The authors propose the incorporation of a nitric oxide donor (SNAP) into a poly-lactide-caprolactone scaffold. They do not study the release of NO from their scaffold, but rely on previous studies using a similar SNAP coating.  The experiment described should be considered as a pilot study, given the low number of animals (low n) used. However, the results they describe are interesting for the tissue engineering community. They find no significant differences between scaffolds with and without SNAP, as to oxygenation, vascularisation and endothelial markers in scaffolds implanted under mouse skin. Their conclusion is sound and well-supported by these data: NO did not improve vascularisation in this model, and the tissue engineering community should be aware that the dose and release kinetics of NO need to be stablished for each biomaterial and implant type.

There are some typographic mistakes that should be corrected:

- line 64: range of action limited to 100-200 unit??.

- line 111: 200 unit?? lectin was injected...

- line 216: a word seems to be missing in the sentence "Previous studies using have shown that.." (using what?).

- references are double-numbered.

Also, the authors should revise the magnification bars in figures 2a and 2b. Figure 2a has a higher magnification than 2b, attending to the size of the vessels and the scaffold. Please, use images of similar magnification.

Author Response

We thank the reviewer for the positive words regarding our manuscript. We hope we have satisfactorily modified the manuscript according to the comments below. 

There are some typographic mistakes that should be corrected:

- line 64: range of action limited to 100-200 unit??.

- line 111: 200 unit?? lectin was injected...

- line 216: a word seems to be missing in the sentence "Previous studies using have shown that.." (using what?).

- references are double-numbered.

We apologize for these mistakes, which mainly occurred when copying the text into the journal template. We carefully revised the whole manuscript to remove these mistakes. 

Also, the authors should revise the magnification bars in figures 2a and 2b. Figure 2a has a higher magnification than 2b, attending to the size of the vessels and the scaffold. Please, use images of similar magnification.

We apologize for the mistake in the magnification bars. We have revised the magnification bars as we, unfortunately, don’t have a high-quality image of similar magnification.